# Research on Macroscopic Mechanical Behavior of Recycled Aggregate Concrete Based on Mesoscale

**DOI:** 10.3390/ma17112532

**Published:** 2024-05-24

**Authors:** Anyu Yang, Qizhi Shang, Yanan Zhang, Junlong Zhu

**Affiliations:** 1Nanjing Hydraulic Research Institute, Nanjing 210024, China; ayyang@hrcshp.org; 2National Research Institute for Rural Electrification of MWR, Hangzhou 310000, China; 3Jiangsu Huaiyin Water Conservancy Construction Co., Ltd., Huaian 223001, China; shangqizhi@126.com; 4College of Water Resources and Hydropower, Hohai University, Nanjing 210024, China; 5School of Engineering Science, University of Science and Technology of China, Hefei 230026, China; zhujunl@mail.ustc.edu.cn

**Keywords:** recycled concrete, regenerated aggregate replacement rate, strain rate, mesoscopic damage evolution, macroscopic performance

## Abstract

Recycled concrete is a heterogeneous composite material, and the composition and volume fraction of each phase affect its macroscopic properties. In this paper, ANSYS APDL was used to construct a two-dimensional numerical model of recycled aggregate concrete with different replacement rates of recycled aggregate (0%, 25%, 50%, 75% and 100%), and a uniaxial compression test was carried out to explore the relationship between recycled aggregate content and its macroscopic mechanical behavior. On this basis, the numerical simulation of different strain rates (0.1 s^−1^, 0.05 s^−1^, 0.01 s^−1^, 0.005 s^−1^ and 0.001 s^−1^) was carried out. It was found that with the increase in the recycled aggregate replacement rate, the peak stress decreases first and then increases, and the peak strain increases continuously. When the replacement rate of recycled aggregate exceeds 50%, the overall damage area of the material increases rapidly. The strain rate will change the path of the micro-crack initiation and expansion of recycled concrete, as well as the process of damage accumulation and evolution. As a result, the unit area and shape of recycled concrete are different at different strain rates, and the damage degree of each phase material is also different.

## 1. Introduction

With the modernization of society, many old buildings generate a large amount of construction waste during demolition. Recycled aggregate concrete (RAC) technology can realize the reprocessing of discarded concrete and restore some of its original performance. RAC is used as a building material in all aspects of human life [1,2,3,4]. Not only can it realize the recycling of construction resources and alleviate the pressure of resource scarcity, but it can also reduce the impact of the construction industry on the environment, which can be said to be a multi-benefit [5,6,7,8]. However, there is a large amount of old mortar in recycled aggregate that cannot be completely removed as it is still attached to the original aggregate particles, and there is a large number of micro-cracks, resulting in the mechanical properties of recycled aggregate and natural aggregate being different.

In recent years, many scholars have conducted a lot of research on the mechanical properties of recycled concrete. Yudong Zhang et al. [9] prepared recycled concrete with two types of recycled aggregates from different sources and investigated the effects of different types of recycled aggregates on recycled concrete; on this basis, the effects of different mineral admixtures on the mechanical properties of recycled concrete were investigated to determine an optimal proportioning scheme, which provided a practical case for the development of recycled concrete. Yuliang Chen et al. [10] explored the mechanical properties of glass fiber recycled concrete (GFRAC) under triaxial compression through conventional triaxial compression tests and established the stress–strain constitutive relationship for glass fiber recycled concrete under triaxial compression based on the test data. Zhang Mingming et al. [11] incorporated burned building material into concrete as a recycled aggregate and found that with the increase in the replacement rate of recycled aggregate, the overall strength, elastic modulus and stress–strain curve of the material gradually decreased, and the brittle failure became more obvious. Yuan Fang et al. [12] proposed an improvement in the ductility of compression-cast RAC through confinement using fiber-reinforced polymer (FRP) laminates and proposed stress–strain models of FRP-confined compression-cast and normal RAC; it was found that the proposed models improved the prediction accuracy of the stress–strain behavior compared with existing models. Yu Yong et al. [13] developed an improved life cycle assessment model to highlight the environmental benefits of using PRAC (recycled aggregate concrete from precast waste) compared to natural aggregate concrete and conventional recycled aggregate concrete. Some key factors influencing PRAC’s environmental performance were also examined. The results indicated that PRAC exhibits around 15% lower energy consumption and carbon dioxide emissions compared to other recycled materials.

At present, most studies on recycled concrete start with macroscopic tests to explore its mechanical properties. Due to the long test period required, significant human and financial resources are required, and the test method is subject to the limitations of the test environment and the original test material, which tend to have high degrees of randomness and many limitations [14,15,16,17]. In addition, as a heterogeneous composite material, the material composition and volume fraction of each phase of recycled concrete have a crucial impact on its mechanical properties, and its macro properties are affected by the micro-structure [18,19,20]. The study of mesoscopic modeling reflects the mechanism of the macroscopic damage of materials more effectively. Therefore, in this paper, two multiphase mesoscopic models of recycled concrete are established at the mesoscopic scale by ANSYS APDL (ANSYS (APDL module): 17.0, ANSYS Inc., Pittsburgh, PA, USA) and equivalization theory, respectively, considering the effect of recycled aggregate content on its macroscopic mechanical behavior and the relationship between its mesoscopic component parameters, and the macroscopic mechanical properties of the material are established. And on this basis, the relationship between the strain rate and material damage evolution is explored, and the macroscopic damage mechanism of the material is investigated at the mesoscopic scale, which provides a theoretical basis for the practical application of recycled concrete. The results of this study can also guide the design of the proportions of materials in recycled concrete materials, promote the development of concrete and provide a sustainable development program for the construction industry.

## 2. Numerical Simulation Scheme

Recycled concrete is composed of recycled materials which can include construction waste, cement admixtures, sand and water. From a mesoscopic point of view, recycled concrete can be regarded as a four-phase composite material composed of a recycled aggregate phase, a new aggregate phase, a cement mortar phase and the interface between the mortar phase and old and new aggregate. In this paper, the meso-model was divided into a regenerative aggregate unit, a new aggregate unit, a cement mortar unit and an interface unit, and the related mechanical numerical calculation was carried out.

### 2.1. Mesoscopic Model

In order to be consistent with the actual recycled concrete test specimens as much as possible, a two-dimensional random polygonal aggregate model was adopted in this paper. The side length of the model was 100 mm × 100 mm, and the particle size ranged from 5 to 20 mm, where the probability $P_{c}(D<D_{0})$ that any point in the two-dimensional plane had an aggregate diameter $D<D_{0}$ was as follows [21,22,23]:(1)$$A_{agg}\left[D_{S},D_{S+1}\right]=\frac{P\left(D_{S+1}\right)-P(D_{S})}{P\left(D_{max}\right)-P(D_{min})}\times R_{agg}\times A_{con}$$
where $A_{agg}$ is the aggregate area with particle size at $\left[D_{S},D_{S+1}\right]$, mm^2^; *P* is the aggregate accumulation distribution function of a particle size; $R_{agg}$ is the area ratio of coarse aggregate, generally 0.4; $A_{con}$ is the area of concrete; in this document, 10,000 mm^2^ is used.

Formula (1) was used to determine the number of aggregates of each particle size in the section of the specimen. According to relevant laboratory test papers [24,25], this paper considered the influence of different recycled aggregate substitutes (0%, 25%, 50%, 75% and 100%) on the mechanical properties of recycled concrete. On this basis, when the replacement rate of recycled aggregate was 100%, the effects of different strain rates of 0.1 s^−1^, 0.05 s^−1^, 0.01 s^−1^, 0.005 s^−1^ and 0.001 s^−1^ on the mechanical properties of recycled concrete were selected, and the meso-model of recycled aggregate with different replacement rates of recycled aggregate was established with the help of the ANSYS APDL (ANSYS (APDL module): 17.0, ANSYS Inc., Pittsburgh, PA, USA) programming language. The specific model is shown in Figure 1.

### 2.2. Failure Criteria

According to the material failure criterion, the maximum tensile stress theory was adopted [26], that is, when the maximum tensile stress of the material exceeds its tensile strength, the material cracks. Combined with the ANSYS life and death element method, when the maximum tensile stress of a material element exceeds its tensile strength, the material element is “killed”; the so-called “kill” does not mean that the element is removed from the model, but the stiffness matrix of the element is multiplied by a very small factor, and the “killed” element load is 0, so as to achieve no effect on the load vector.

### 2.3. Material Parameter

In order to avoid repeated experiments, the numerical value adopted in this study was calculated by referring to other people’s existing research results [27,28], as shown in Table 1.

## 3. Simulation Results and Discussion

### 3.1. Effects of Different Aggregate Substitution Rates

#### 3.1.1. Stress–Strain Relationship

Through the numerical analysis of the failure process of RA under five conditions of R = 0%, 25%, 50%, 75% and 100% (the strain rate was 0.02 s^−1^) with the recycled aggregate replacement rate, the stress and deformation of the test block under each load step could be obtained, as shown in Figure 2 below.

As can be seen from Figure 2, with the increase in the replacement rate of recycled aggregate, the corresponding peak stress point gradually decreased. The reason for this phenomenon is that the strength of recycled aggregate is lower than that of natural aggregate, and as the content of recycled aggregate increases, the overall strength of recycled concrete specimens decreases. At the same time, with the increase in the replacement rate of recycled aggregate, the strain size of the specimen when it reached the peak point was also increasing, and the curve as a whole showed a trend of right–downward deviation. As the elastic modulus of recycled aggregate was lower than that of natural aggregate, the overall elastic modulus of the specimen was constantly lower when the replacement rate of recycled aggregate was high, which would make the curve flatter. On the whole, the stress–strain curves obtained under the five conditions with different substitution rates all had similar curve forms, which is consistent with the conclusion obtained in the actual test [29], which proves the rationality of the model construction. By comparison, it was found that the peak strain of each curve showed a trend to the right as a whole, which indicates that the recycled aggregate contained a large amount of waste cement slurry, while the old cement slurry had more holes and cracks, resulting in a large change in the peak value when the recycled concrete was damaged.

#### 3.1.2. Mesoscopic Damage Evolution

Uniaxial compression simulation was carried out on the numerical model of RAC with five different aggregate replacement rates of R = 0%, 25%, 50%, 75% and 100%, and the failure process of loading to four states of failure generation, failure development, failure intensification and failure completion was recorded. See Figure 3 for details. The white unit in the figure is the killed unit, which is the area where the damage occurred. The first principal stresses in the specimen at the time of damage generation and development are specified in Figure 4 below.

In Figure 3 and Figure 4, it can be seen that due to the phenomenon of stress concentration and the low value of interface strength, the initial failure area of the material was at the interface of the corner point of large aggregate and small aggregate in the middle, especially at the convex angle of the aggregate. Since the strength value of natural aggregate is higher than that of recycled aggregate, in the working condition R = 100%, the area in the middle of the specimen except aggregate was basically destroyed, and with the increase in stress, the cement mortar unit needed to bear more stress, so the damage was more thorough.

The kill unit area of each load step under the above five conditions was extracted and treated, and the evolution process of the mesoscopic damage area along with the load step under different regenerated aggregate replacement rates was determined, as shown in Figure 5 below. The number of steps at the time of failure, the final area of failure and the total load steps used to complete the failure were recorded. At the same time, the change in the final area of failure at different replacement rates of recycled aggregates was calculated based on the 0% replacement rate condition, that is, the condition that all aggregates were natural aggregates. The recording and calculation process are shown in Table 2.

It can be seen from Figure 5 and Table 2 that (1) the destruction of the material was a process of embryo formation, inoculation, expansion and convergence of the internal microdefects. Therefore, with the increase in the number of load steps, the growth rate of the damaged area changed from slow to fast and eventually tended to a fixed value. (2) In the meso-structure of recycled concrete, the property of the recycled aggregate phase unit is weaker than that of natural aggregate phase, and the strength value of the interface area attached around it is also weaker than or equal to the cement mortar phase, which is more likely to cause damage. Therefore, when R = 0%, 25% or 50%, the whole recycled concrete material was still supported by natural aggregate, and the bearing effect of recycled aggregate was not obvious. After the failure of recycled aggregate, the whole specimen would have been completely destroyed, and the damaged area would have no longer increased, resulting in the final damaged area decreasing. When R = 75% or 100%, the whole material was supported by recycled aggregate, the phase strength of recycled aggregate was not much different from that of cement mortar and the specimen was stressed evenly. Compared with the specimen with a low recycled aggregate replacement rate, the specimen began to fail late, but with the gradual increase in stress, the final failure area would have become larger and larger, and the damage would have been the most severe. (3) Recycled aggregate has a lower elastic modulus than natural aggregate, and when the same deformation occurs with natural aggregate, recycled aggregate bears less stress, resulting in a later failure time, which also shows that recycled aggregate has a certain weakening effect on stress concentration. Therefore, with the increase in the regenerated aggregate replacement rate, the number of load steps during the failure of the specimen also increased.

### 3.2. Effects of Different Strain Rates

#### 3.2.1. Stress–Strain Relationship

A mesoscopic model with R = 100% was selected to carry out the numerical simulation of uniaxial compression at different strain rates of 0.1 s^−1^, 0.05 s^−1^, 0.01 s^−1^, 0.005 s^−1^ and 0.001 s^−1^. The stress–strain relationship is shown in Figure 6 below.

It can be seen from Figure 6 that (1) the strain rate did not change the stress–strain curve shape of the material; the higher the strain rate, the greater the peak stress, the larger the elastic modulus value and the approximate linear growth, which was consistent with the actual test results [30]. (2) In the numerical simulation calculation, the strain was accumulated in the failure unit grid standard during the process of stress bearing of the material, while in the actual laboratory test, the strain rate was too fast, which would have led to the beginning of new failure before the complete extension of the failure in the previous stage. Therefore, the increase in strain rate in the numerical calculation made the peak strain result of the RAC specimen larger.

#### 3.2.2. Mesoscopic Damage Evolution

The failure process of recycled concrete under different strain rates in four states, failure generation, failure development, failure intensification and failure completion, was determined. See Figure 7 for details. The white part in the figure is the killed unit, that is, the area where the failure occurs.

As can be seen from Figure 7, (1) the strain rate will change the path, mode and cumulative evolution process of micro-crack initiation and expansion in recycled concrete, resulting in different areas and shapes when the material is finally damaged. (2) Under the condition of low strain rate, the development of failure shape was slow, the final failure shape was mostly transverse penetration, the failure area was small, the failure area was mostly an interface unit and cement mortar unit and the damage to the aggregate unit was low. At high strain rates, the failure shape developed rapidly, and the final failure shape was diagonal along the upper right to the lower left corner. The internal failure area was large, and most of the aggregate units also produced failure.

## 4. Conclusions

In order to explore the effects of recycled aggregate content and strain rate on the mechanical properties and mesoscopic damage mechanism of RAC, ANSYS APDL was used to construct two-dimensional finite-element models of RAC under different working conditions, and numerical simulation calculation of the uniaxial compression experiment was carried out. The failure phenomenon of the material was consistent with that of the laboratory test, which verified the correctness of the numerical simulation. The research found the following:(1)With the increase in the replacement rate of recycled aggregate, the peak stress first decreases and then increases, and the peak strain keeps increasing. Since the overall stress of the material depends on the balance of the aggregate strength, the closer the aggregate strength is, the greater the overall strength of the material will be, and the greater the deviation of the aggregate strength will be, so the overall strength of the material will be affected by the low aggregate strength.(2)The interfacial transition area between recycled aggregate and cement slurry is the first place where the material begins to fail, among which, the interfacial area at the corner point of the aggregate is the most prone to failure due to the phenomenon of stress concentration. With the increase in the regenerated aggregate replacement rate, the area of the final damaged region will also increase. When the regenerated aggregate replacement rate exceeds 50%, the overall damaged area of the material will be larger due to the brittle condition of the aggregate.(3)Strain rate will change the path of the micro-crack initiation and expansion of recycled concrete, as well as the process of damage accumulation and evolution, resulting in different areas and shapes of recycled concrete when the final failure is completed at different strain rates and different damage degrees of materials in different phases.

## Figures and Tables

**Figure 1 materials-17-02532-f001:**
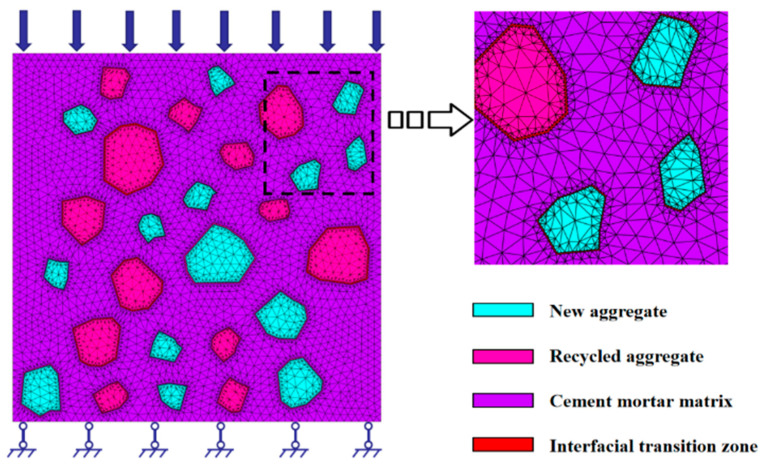
Meso-mechanical model of RAC.

**Figure 2 materials-17-02532-f002:**
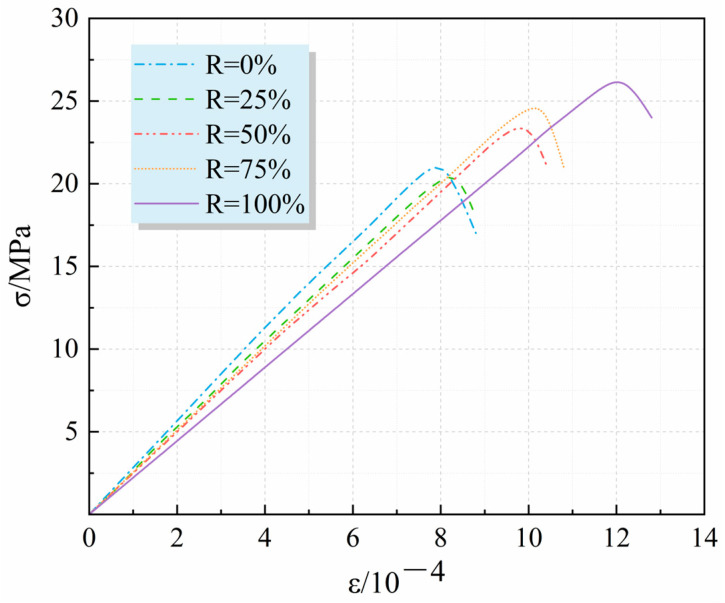
Stress–strain curves under different replacement rates of recycled aggregate.

**Figure 3 materials-17-02532-f003:**
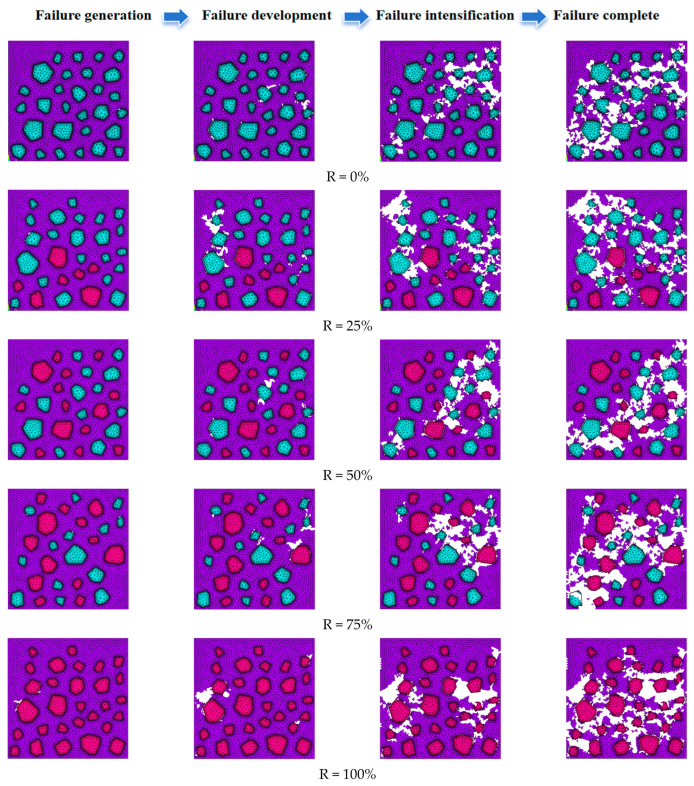
Mesoscopic failure process of RAC with different aggregate replacement rates.

**Figure 4 materials-17-02532-f004:**
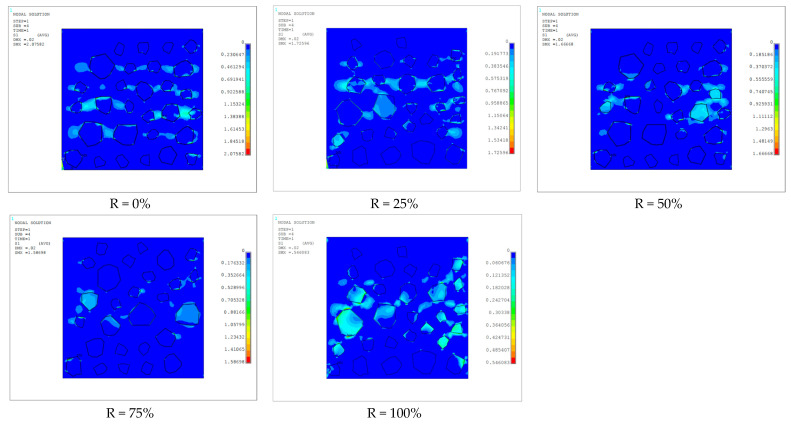
First principal stress.

**Figure 5 materials-17-02532-f005:**
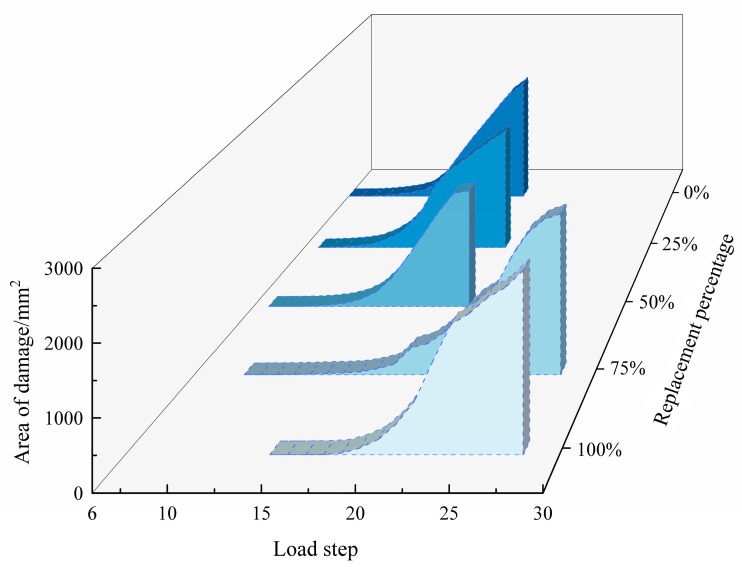
Changes in mesoscopic damage area under different working conditions.

**Figure 6 materials-17-02532-f006:**
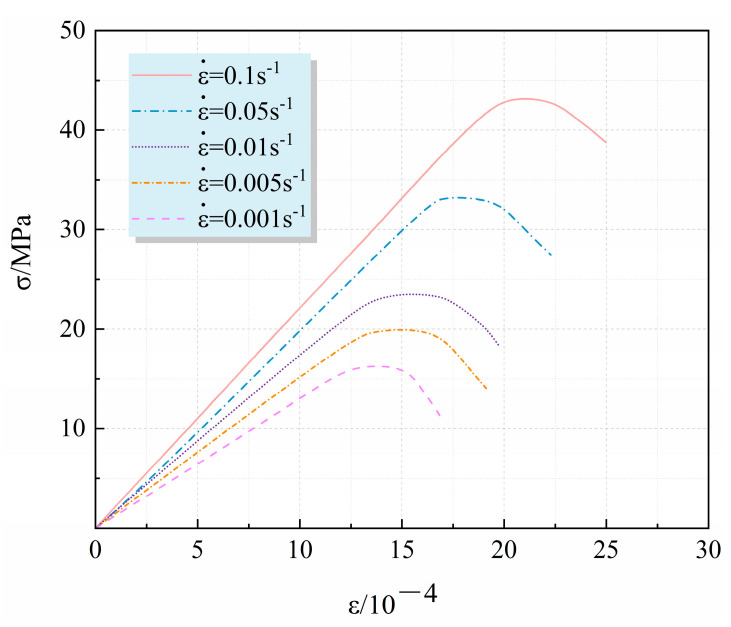
Stress–strain curves at different strain rates (R = 100%).

**Figure 7 materials-17-02532-f007:**
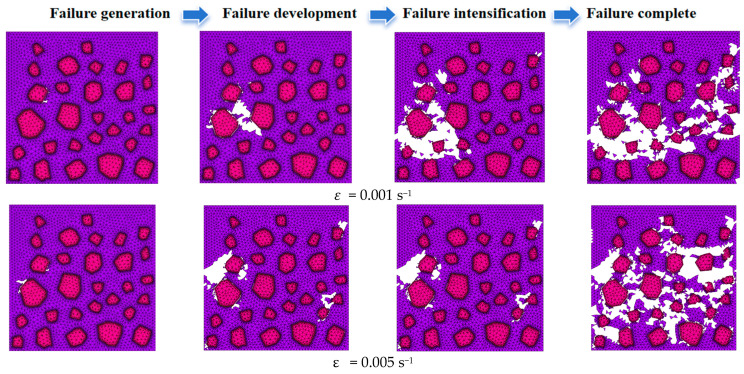
Mesoscopic failure process of RAC at different strain rates.

**Table 1 materials-17-02532-t001:** Parameters of each mesoscopic component of RAC.

Mesoscopic Component	Elasticity Modulus/MPa	Poisson’s Ratio	Tensile Strength/MPa
Natural aggregate	70	0.16	10
Recycled aggregate	32	0.167	3.2
Cement mortar	25	0.2	3.0
Interfacial transition zone	14	0.22	1.4

**Table 2 materials-17-02532-t002:** Failure change data under each working condition.

Regenerated Aggregate Replacement Rate	Failure Load Beginning Steps	Failure Load Completion Steps	Final Damaged Area/mm^2^	Final Destruction Area Growth Rate
0%	5	14	2100.18	0%
25%	6	14	2024.69	−3.59%
50%	6	13	1885.41	−10.23%
75%	8	20	2426.41	15.53%
100%	13	15	2560.81	21.93%

## Data Availability

The original contributions presented in the study are included in the article, further inquiries can be directed to the corresponding author.

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
