# Peer review of "Research on Macroscopic Mechanical Behavior of Recycled Aggregate Concrete Based on Mesoscale"

_materials, 2024, doi:10.3390/ma17112532_

Round 1

Reviewer 1 Report

Comments and Suggestions for Authors

The work is theoretical in nature and concerns the broadly understood mechanics of materials. The authors analyzed the behavior of a concrete model made with the addition of recycled aggregate subjected to axial compression. The authors used the ANSYS APDL system and built a 2D model of concrete with the addition of 0%, 25%, 50%, 75% and 100% recycled aggregate. The authors analyzed the macroscopic behavior in the form of damage development and the stress-strain relationship. The analyzes were performed at various strain rates from 0.1s-1 to 0.001s-1. It has been shown that an increase in aggregate content causes an increase in maximum stresses. At the same time, the overall area of ​​material damage reaches a maximum at approximately 50% recycled aggregate content. In turn, the rate of strain application changes the initiation site and the damage field. Taking into account the content of recycled aggregate, a different mode of damage to concrete phases is obtained.

I evaluate the work positively because it concerns the interesting problem of reusing concrete from demolition works. There are certain issues that are necessary for the work to be accepted for publication. I present my detailed comments below.

1. Chapter 1. In the analytical part, very little space is devoted to numerical modeling and material models of concrete. Of course, material micromodels were mentioned, but there is no description of the model in the ANSYS Willam-Warnke system. what are the most important differences in the macromodeling technique used. It is proposed to emphasize the contribution of the publication to the development of the discipline of materials engineering.

2. Chapter 2.1. Please explain whether the concrete model used corresponds to real concrete with the addition of recycled aggregate - whether the aggregate stack used with a grain diameter >2 mm is identical to that in concrete. Why was the elastic-brittle Rankine model used for mortar? The mortar has elastic-plastic properties, especially with slowly changing deformations. What were the dimensions of the analyzed sample and why was the stocky model chosen?

3. Chapter 2.3. The given parameters of the materials used require better explanation - especially the methodology for determining the parameters.

4. Chapter 3.1.2. The presented scratch images do not correspond to the classic mechanism of damage to stocky concrete samples in which the formation of two truncated pyramids is observed - how to explain it. How do the authors explain the destruction of concrete, basically only within the mortar? I think it is necessary to show, in addition to the damage development, also the results in the form of principal stresses.

5. Chapter 3.2.2. Evidently, the damage depends on the strain rate, and if so, which of the material parameters was responsible for such an effect - the dynamic modulus of elasticity? This issue requires clarification.

6. Chapter 4. I propose to add information in the conclusions that the obtained results must be confirmed empirically. Models should be calibrated.

Author Response

Dear Editors and Reviewers:

Thank you for your letter and for the reviewers' comments our manuscript entitled “Research on Macroscopic Mechanical Behavior of Recycled Aggregate Concrete Based on Mesoscale”(materials-2993065).

Frist, We extend our heartfelt thanks to reviewers and editors for their valuable comments and suggestions. Those comments are all valuable and very helpful for revising and improving our paper, as well as the important guiding significance to our researches. We have studied comments carefully and have made correction which we hope meet with approval. Revised portion are marked in yellow in the paper. The main corrections in the paper and the responds to the reviewer's comments are as flowing:

The work is theoretical in nature and concerns the broadly understood mechanics of materials. The authors analyzed the behavior of a concrete model made with the addition of recycled aggregate subjected to axial compression. The authors used the ANSYS APDL system and built a 2D model of concrete with the addition of 0%, 25%, 50%, 75% and 100% recycled aggregate. The authors analyzed the macroscopic behavior in the form of damage development and the stress-strain relationship. The analyzes were performed at various strain rates from 0.1s-1 to 0.001s-1. It has been shown that an increase in aggregate content causes an increase in maximum stresses. At the same time, the overall area of material damage reaches a maximum at approximately 50% recycled aggregate content. In turn, the rate of strain application changes the initiation site and the damage field. Taking into account the content of recycled aggregate, a different mode of damage to concrete phases is obtained. I evaluate the work positively because it concerns the interesting problem of reusing concrete from demolition works. There are certain issues that are necessary for the work to be accepted for publication. I present my detailed comments below.

1. Chapter 1. In the analytical part, very little space is devoted to numerical modeling and material models of concrete. Of course, material micromodels were mentioned, but there is no description of the model in the ANSYS Willam-Warnke system. what are the most important differences in the macro modeling technique used. It is proposed to emphasize the contribution of the publication to the development of the discipline of materials engineering.

Response:
(1) Models from ANSYS have been included upon request.
(2) Macro-modelling is usually an in-house experimental means to carry out the process of recycled concrete mixing, curing, and testing to explore the overall damage of the material; compared with mesoscopic modelling, which has a long test period and requires manpower and material resources to complete.
(3) its contribution to the development of the discipline of materials engineering has been included in the text.

2. Chapter 2.1. Please explain whether the concrete model used corresponds to real concrete with the addition of recycled aggregate - whether the aggregate stack used with a grain diameter >2 mm is identical to that in concrete. Why was the elastic-brittle Rankine model used for mortar? The mortar has elastic-plastic properties, especially with slowly changing deformations. What were the dimensions of the analyzed sample and why was the stocky model chosen?

Response:
(1)Consistent with the recycled concrete aggregate is crushed, the shape is basically convex, so it can be simplified into an irregular polygon; uniaxial compression test of recycled concrete is usually used for the side length of 100mm or 150mm cube, taking into account the complexity of the calculation of the three-dimensional model, the two-dimensional model is more simple, so this ANSYS simulation of the model shape of the side of the length of 100mm * 100mm, the range of the random aggregate particle size of 5 ~ 20mm;
(2)Generally speaking, the material will have large non-linear deformation before fracture, but for RAC material, it is brittle material, and the deformation is very small. The nonlinear behavior of materials is very important, but it is difficult to obtain and verify this complex behavior by means of experiment or numerical simulation, especially the mesoscopic components of composite materials. In this paper, the linear elasticity of aggregate, interface and other mesoscopic components is taken into account from the mesoscopic perspective of RAC material, and the failure criterion is taken into account. From the macro perspective, RAC composite material is nonlinear, and simple model parameters are convenient for engineering application.

3. Chapter 2.3. The given parameters of the materials used require better explanation - especially the methodology for determining the parameters.

Response: Recycled concrete is divided into recycled aggregate phase, natural aggregate phase, cement mortar phase and interface phase from mesoscopic scale, and it is difficult to obtain the mesoscopic parameters through relevant experiments, therefore, through relevant literature research, previous scholars have determined the relevant parameters through experiments or books. From the paper ‘Damage and Destruction Research of Recycled Concrete with Waste Brick Based on Modified Random Aggregate Model’, the fine parameters of recycled aggregate phase, cement mortar phase and interface phase can be determined, and from the paper ‘Meso-level Numerical Simulation on Mechanical Properties of Modeled Recycled Concrete Under Uniaxial Compression’, the mesoscopic parameters of natural aggregate phase can be determined.

4. Chapter 3.1.2. The presented scratch images do not correspond to the classic mechanism of damage to stocky concrete samples in which the formation of two truncated pyramids is observed - how to explain it. How do the authors explain the destruction of concrete, basically only within the mortar? I think it is necessary to show, in addition to the damage development, also the results in the form of principal stresses.

Response:
(1) uniaxial pressure due to constraints, the test block in the middle of the bulge, the middle of the test block near the boundary of the region of the first principal stress is larger, the region of the unit is first destroyed; the reason for this phenomenon is that the strength of the aggregate and mortar are different, the concentration of stress is mainly generated in the particles between the particles, especially in the convex corners of the aggregate; with the increasing pressure, the small cracks will gradually expand, extend, through the mortar , forming a form of damage around the aggregate;
(2) In this study, it is set that if the aggregate of recycled concrete material is damaged, the material is completely destroyed, so the results after the destruction of the aggregate are not studied, and only the process before the destruction of the aggregate is considered;
(3) The result of principal stress has been added.

5. Chapter 3.2.2. Evidently, the damage depends on the strain rate, and if so, which of the material parameters was responsible for such an effect - the dynamic modulus of elasticity? This issue requires clarification.

Response: This effect is due to the fact that at high strain rates, the initial ‘stiffness’ of the recycled concrete material increases and reaches the critical state earlier, resulting in a lower deformability and poorer ductility, i.e., a higher modulus of elasticity.

6. Chapter 4. I propose to add information in the conclusions that the obtained results must be confirmed empirically. Models should be calibrated.

Response: Added to the conclusion of the article

Reviewer 2 Report

Comments and Suggestions for Authors

The research presented by the authors is interesting as we need accurate computer models to reduce the amount of laboratory works and generation of wastes.

However, the authors are kindly required to consider the following comments during the revision process of their manuscript.

Title - is somehow incomplete such as "microscale...?" Moreover, there is no hit that only numerical simulations were used. 

Line 21 - what area? recycled concrete in general or the geometry of the recycled concrete sample?

Lines 72 - 78 - the sentence is too long. Additionally, only parts of the proposed goals are presented in the manuscript, sometimes a bit unclear. Last but not least, the aim of the manuscript, as stated in here, is only remotely related to the title itself.

Section 2.1 - is very short and only scarce information is given, considering one of the main aims of the manuscript as highlighted at lines 72-78. It is unclear what was the maximum aggregate size considered in the model, the dimensions of the model (several times larger than the max aggregate size, as seen from Figure 1).

Line 93 - based on what was the 0.4 value selected. There is no justification of this number either through references or, preferably, thorough justification with references included. Acon is a very important parameter and should be specified because it leads to the max aggregate size selection and influences all other parameters in equation 1.

Line 100 - it is here that the reader finds out about the micro-model (missing from the title). Lines 72-78 state the use of mesoscopic model, without any further explanations.

Section 2.3, line 115 - reference 25 is cited as the source of material characteristics, yet reference 25 is related to bricks (albeit using recycled concrete aggregates). Why was this source selected and not the ones from references 26 and 27, used to compare your results with?

Line 117 - what is the dielectric material used in this study?

Figure 2 - what was the strain rate used for obtaining those graphs? Was the same phenomenon observed for all strain rates?

Lines 139-141 - this is a behavior associated to any type of concrete, not only RAC. It is general knowledge so it doesn't need to be repeated here.

Lines 143-145 - without any numbers, the statement is mostly speculation. It would be advisable to correlate your data with other reported results from the scientific literature (preferably international).

Figure 3 - what are the 4 stages for each considered mix corresponding to the graphs in Figure 2? Probably, each figure was considered at a certain strain-stress value?

Figure 4 - load step number is rather irrelevant. A more useful information would be the strain or the stress value. Please change axis title from "working condition" to "replacement percentage"

Table 2 - similar comment to Figure 4. Analysis step numbers are irrelevant.

Section 3.2.1. - why are the results shown only for 100% replacement? Were all other replacement percentages leading to similar results? If so, please specify in the text.

Line 214 - could you confirm with reference 26, too?

Figure 6 - the same RAC as in Figure 5?

Comments on the Quality of English Language

Line 16 and throughout the manuscript - what do you understand by regenerated aggregate? Are the authors referring to recycled aggregates? If so, please use the latter formulation.

Line 28 - the dismantling and rebuilding processes have different goals, not to produce construction wastes. Please rephrase.

Line 31 - "it can not only ....recycled" is not clear. Please rephrase.

Line 33 - what is "plurality of birds"? please substitute "number" by "amount / quantity / volume"

Lines 33 - 36 - the sentence is too long. Please split it into shorter sentences.

Lines 38 - 42 - the sentence is too long. Please split it into shorter sentences.

Lines 47 - 53 - it is difficult to understand what the authors are trying to say. Please rephrase and use shorter sentences.

Line 56 - the sentence should end at RAC

Lines 153-162 (174-200; 208-219; 226-235) - please use narration to explain the findings, not enumeration.

Line 160 - damage is more extended?

Line 170 - are shown

Line 252 - please substitute "area" by "region" to avoid unnecessary repetitions

Author Response

Dear Editors and Reviewers:

Thank you for your letter and for the reviewers' comments our manuscript entitled “Research on Macroscopic Mechanical Behavior of Recycled Aggregate Concrete Based on Mesoscale”(materials-2993065). Frist, We extend our heartfelt thanks to reviewers and editors for their valuable comments and suggestions. Those comments are all valuable and very helpful for revising and improving our paper, as well as the important guiding significance to our researches. We have studied comments carefully and have made correction which we hope meet with approval. Revised portion are marked in yellow in the paper. The main corrections in the paper and the responds to the reviewer's comments are as flowing:

The research presented by the authors is interesting as we need accurate computer models to reduce the amount of laboratory works and generation of wastes.

However, the authors are kindly required to consider the following comments during the revision process of their manuscript.

Title - is somehow incomplete such as "microscale...?"  Moreover, there is no hit that only numerical simulations were used.

Response: Written incorrectly, it should be mesoscale; the title does not mention the means of the study because if numerical simulations were added, the title would be too long and would not be a major part of the study.

Line 21 - what area?  recycled concrete in general or the geometry of the recycled concrete sample?

Response: It refers to the area and shape of the destructive unit of recycled concrete during compression, which has been modified in the article.

Lines 72 - 78 - the sentence is too long.  Additionally, only parts of the proposed goals are presented in the manuscript, sometimes a bit unclear.  Last but not least, the aim of the manuscript, as stated in here, is only remotely related to the title itself.

Response: Thanks for raising the issue, changes have been made in the article.

Section 2.1 - is very short and only scarce information is given, considering one of the main aims of the manuscript as highlighted at lines 72-78.  It is unclear what was the maximum aggregate size considered in the model, the dimensions of the model (several times larger than the max aggregate size, as seen from Figure 1).

Response: Thanks for raising the issue, changes have been made in the article.

Line 93 - based on what was the 0.4 value selected.  There is no justification of this number either through references or, preferably, thorough justification with references included.  Acon is a very important parameter and should be specified because it leads to the max aggregate size selection and influences all other parameters in equation 1.

Response: References have been provided and the parameters of Acon are given in the text.

Reference: [1]Wang Z M , Kwan A , Chan H C .Mesoscopic study of concrete I: generation of random aggregate structure and finite element mesh[J].Computers & Structures,1999, 70:533-544.

[2]Liu T ,Qin S ,Zou D , et al. Mesoscopic modeling method of concrete based on statistical analysis of CT images[J].Construction and Building Materials,2018,192429-441.

Line 100 - it is here that the reader finds out about the micro-model (missing from the title).  Lines 72-78 state the use of mesoscopic model, without any further explanations.

Response: The wording in the article is inaccurate, here it should be meso-modelling, which has been amended.

Section 2.3, line 115 - reference 25 is cited as the source of material characteristics, yet reference 25 is related to bricks (albeit using recycled concrete aggregates).  Why was this source selected and not the ones from references 26 and 27, used to compare your results with?

Response: References 26 and 27 deal only with the macroscopic findings and do not mention the determination of the mesoscopic component parameters, for which the literature source has been supplemented with another paper, and the parameters were determined based on the two papers together. From the paper ‘Damage and Destruction Research of Recycled Concrete with Waste Brick Based on Modified Random Aggregate Model’, the fine parameters of recycled aggregate phase, cement mortar phase and interface phase can be determined, and from the paper ‘Meso-level Numerical Simulation on Mechanical Properties of Modeled Recycled Concrete Under Uniaxial Compression’, the mesoscopic parameters of natural aggregate phase can be determined.

Line 117 - what is the dielectric material used in this study?

Response: The writing was incorrect and has been corrected.

Figure 2 - what was the strain rate used for obtaining those graphs?  Was the same phenomenon observed for all strain rates?

Response: The strain rate in Fig. 2 is 0.02s-1. The study of the effect of strain rate on the material is mainly in section 3.2 of the article, where specific problems are analysed.

Lines 139-141 - this is a behavior associated to any type of concrete, not only RAC.  It is general knowledge so it doesn't need to be repeated here.

Response: Thanks, it has been deleted.

Lines 143-145 - without any numbers, the statement is mostly speculation.  It would be advisable to correlate your data with other reported results from the scientific literature (preferably international).

Response: Numerous literature and reports have been searched and there is no similar.

Figure 3 - what are the 4 stages for each considered mix corresponding to the graphs in Figure 2?  Probably, each figure was considered at a certain strain-stress value?

Response: The four stages in Figure 3 all fall before the peak of the stress-strain curve and were selected based on material damage and not based on stress or strain values, as Figure 3 mainly analyses the material damage process. Generally speaking, the material will have large non-linear deformation before fracture, but for RAC material, it is brittle material, and the deformation is very small. The nonlinear behavior of materials is very important, but it is difficult to obtain and verify this complex behavior by means of experiment or numerical simulation, especially the mesoscopic components of composite materials.

Figure 4 - load step number is rather irrelevant.  A more useful information would be the strain or the stress value.  Please change axis title from "working condition" to "replacement percentage"

Table 2 - similar comment to Figure 4.  Analysis step numbers are irrelevant.

Response: (1) This idea was not considered at the beginning of the study, and the results were obtained without determining its corresponding stress or strain values, if considered, the calculations would need to be repeated again, which is a bit too much work, and it is planned to consider the idea in the next study, thank you very much for your suggestion. (2) Figure 4 has been modified.

Section 3.2.1.  - why are the results shown only for 100% replacement?  Were all other replacement percentages leading to similar results?  If so, please specify in the text.

Response: Due to the heavy workload, only R=100% has been considered so far, and the results of other working conditions need to be analyzed more specifically, which is what we are going to do in the next step.

Line 214 - could you confirm with reference 26, too?

Response: Yes, because a relevant study was carried out in reference 26 by means of an indoor experiment, and its comparison with it can verify the feasibility of the research model in this paper.

Figure 6 - the same RAC as in Figure 5?

Response: Yes, the same.

Comments on the Quality of English Language

Line 16 and throughout the manuscript - what do you understand by regenerated aggregate?  Are the authors referring to recycled aggregates?  If so, please use the latter formulation.

Line 28 - the dismantling and rebuilding processes have different goals, not to produce construction wastes.  Please rephrase.

Line 31 - "it can not only .... recycled" is not clear.  Please rephrase.

Line 33 - what is "plurality of birds"?  please substitute "number" by "amount / quantity / volume"

Lines 33 - 36 - the sentence is too long.  Please split it into shorter sentences.

Lines 38 - 42 - the sentence is too long.  Please split it into shorter sentences.

Lines 47 - 53 - it is difficult to understand what the authors are trying to say.  Please rephrase and use shorter sentences.

Line 56 - the sentence should end at RAC

Lines 153-162 (174-200;  208-219;  226-235) - please use narration to explain the findings, not enumeration.

Line 160 - damage is more extended?

Line 170 - are shown

Line 252 - please substitute "area" by "region" to avoid unnecessary repetitions

Response: It has been modified.

Round 2

Reviewer 1 Report

Comments and Suggestions for Authors

I am reviewing the work for the second time. Compared to the first version, a number of changes and additions have been made. I respond to my detailed comments below.

1. Chapter 1. In the analytical part, very little space is devoted to numerical modeling and material models of concrete. Of course, material micromodels were mentioned, but there is no description of the model in the ANSYS Willam-Warnke system. what are the most important differences in the macromodeling technique used. It is proposed to emphasize the contribution of the publication to the development of the discipline of materials engineering.

Additions were made and my doubts were clarified. I have no further comments.

2. Chapter 2.1. Please explain whether the concrete model used corresponds to real concrete with the addition of recycled aggregate - whether the aggregate stack used with a grain diameter >2 mm is identical to that in concrete. Why was the elastic-brittle Rankine model used for mortar? The mortar has elastic-plastic properties, especially with slowly changing deformations. What were the dimensions of the analyzed sample and why was the stocky model chosen?

My doubts were clarified. I have no comments.

3. Chapter 2.3. The given parameters of the materials used require better explanation - especially the methodology for determining the parameters.

Additions have been made. I have no comments.

4. Chapter 3.1.2. The presented scratch images do not correspond to the classic mechanism of damage to stocky concrete samples in which the formation of two truncated pyramids is observed - how to explain it. How do the authors explain the destruction of concrete, basically only within the mortar? I think it is necessary to show, in addition to the damage development, also the results in the form of principal stresses.

Added additions. I have no comments.

5. Chapter 3.2.2. Evidently, the damage depends on the strain rate, and if so, which of the material parameters was responsible for such an effect - the dynamic modulus of elasticity? This issue requires clarification.

Clarified and supplemented. I have no comments.

6. Chapter 4. I propose to add information in the conclusions that the obtained results must be confirmed empirically. Models should be calibrated.

Added addition. I have no comments.

Author Response

Dear Editor and reviewer,

The format of the paper references has been revised.

Reviewer 2 Report

Comments and Suggestions for Authors

The authors addressed all comments raised during the review process. Additional explanations were offered which helped improving the quality of the manuscript.

I, therefore, endorse this manuscript for publication.

Author Response

(The authors gave the same response as above.)
